# Potential Impact of Preoperative Circulating Biomarkers on Individual Escalating/de-Escalating Strategies in Early Breast Cancer

**DOI:** 10.3390/cancers15010096

**Published:** 2022-12-23

**Authors:** Caterina Gianni, Michela Palleschi, Filippo Merloni, Sara Bleve, Chiara Casadei, Marianna Sirico, Giandomenico Di Menna, Samanta Sarti, Lorenzo Cecconetto, Marita Mariotti, Ugo De Giorgi

**Affiliations:** Department of Medical Oncology, IRCCS Istituto Romagnolo per lo Studio dei Tumori (IRST) “Dino Amadori”, 47014 Meldola, Italy

**Keywords:** biomarkers, early breast cancer, ctDNA, cfDNA, fragmentomic, miRNA, methylation, neoadjuvant therapy, escalation, de-escalation

## Abstract

**Simple Summary:**

Advancements in molecular technologies have extended the potential role of circulating biomarkers in diagnosis, disease monitoring, and outcome prediction. Research is working towards personalized treatment in early breast cancer patients, and tools for intensification/de-intensification strategies are needed to guide clinicians’ choices. In this review, we explore the current developments of promising biomarkers in early breast cancer for cancer detection and tailoring neoadjuvant therapies.

**Abstract:**

The research on non-invasive circulating biomarkers to guide clinical decision is in wide expansion, including the earliest disease settings. Several new intensification/de-intensification strategies are approaching clinical practice, personalizing the treatment for each patient. Moreover, liquid biopsy is revealing its potential with multiple techniques and studies available on circulating biomarkers in the preoperative phase. Inflammatory circulating cells, circulating tumor cells (CTCs), cell-free DNA (cfDNA), circulating tumor DNA (ctDNA), and other biological biomarkers are improving the armamentarium for treatment selection. Defining the escalation and de-escalation of treatments is a mainstay of personalized medicine in early breast cancer. In this review, we delineate the studies investigating the possible application of these non-invasive tools to give a more enlightened approach to escalating/de-escalating strategies in early breast cancer.

## 1. Introduction

Early breast cancer (eBC) is considered a potentially curable disease; however, nearly 30% of cases will evolve into metastatic disease even after a long follow-up [1]. Defining the escalation and de-escalation of treatments is a mainstay of personalized medicine in eBC to guarantee the best results in terms of outcome and acceptable toxicities. The BC prognosis depends on multiple biological factors regarding the tumor and individual characteristics; however, patients with similar biological factors often experience different survival outcomes, and currently, there is no answer to these disparities. The ongoing research is trying to discover new biomarkers to better estimate patients’ survival outcomes and to identify subjects with a high risk of relapse that may deserve a strengthened treatment approach [2,3]. In the era of precision medicine, non-invasive methods to better stratify patients are in constant development to complement the predictive and prognostic tools currently used in the clinic.

Liquid biopsy is the most intriguing resource for circulating biomarkers such as cell-free DNA, circulating tumor DNA (ctDNA), circulating tumor cells (CTCs), and circulating tumor RNA (ctRNA). Furthermore, the peripheral inflammatory cells may easily be evaluated as non-invasive biomarkers from a simple blood sample, revealing information even at the early stages of the diagnostic–therapeutic approach [4]. Other information may be extracted from the analysis of circulating cfDNA fragments and epigenomic analysis [5]. Early cancer detection with liquid biopsy is a new challenge, and prospective studies are currently ongoing [6,7,8].

In the preoperative setting of eBC, individualizing predictive response factors to neoadjuvant treatment and the achievement of a pathological complete response (pCR) is still an unmet need. PCR is increasingly being considered as a potential surrogate for survival outcomes, but there is still a lack of evidence to support its connection with different survival endpoints [9]. As a consequence, Conforti and colleagues suggest in a large meta-analysis that pCR may not be used as the primary endpoint in clinical trials [9]. However, pCR is a determinant in the choice of sequential treatments. In particular, regarding human epidermal growth factor receptor 2 positive (HER-2+) eBC, the NeoSphere randomized phase 2 trial showed some benefit in terms of pCR in operable, locally advanced, and inflammatory BC when treated with four neoadjuvant cycles of docetaxel/pertuzumab/trastuzumab and also demonstrated favorable progression-free survival outcomes at the 5-year follow-up [10]. Consequently, the achievement of pCR affects the adjuvant treatment options in HER-2+ eBC patients. According to the results from the Katherine trial, patients with residual invasive disease benefitted from trastuzumab emtansine as adjuvant therapy after a dual-block neoadjuvant treatment [11].

Similarly, in triple-negative (TN) disease, the use of neoadjuvant pembrolizumab in association with standard chemotherapy has been shown to increase the possibility of pCR achievement [12]. However, it is not yet clear which subgroup of eBC patients may benefit from intensive neoadjuvant approaches; in many cases, some patients may not derive any outcome benefit but may experience more toxicities (sometimes irreversible). Therefore, not all subtypes are equal to the others. As demonstrated, pCR is not considered a suitable surrogate endpoint, especially for patients with luminal B/HER2-positive, luminal A tumors, or lobular subtypes [13].

In the view of personalized care, it is clear there is a need for useful tools to predict treatment efficacy and prognosis before surgery and presurgical treatments to improve the therapeutic strategy. New algorithms have been developed to extrapolate the prognostic and predictive information provided by tissue biomarkers such as genomic tests prospectively validated in the adjuvant setting (such as Oncotype DX, Prosigna, and Mammaprint), and others are under development (such as the HER2DX test also used at the preoperative stage) [2,14]. In particular, HER2DX is a new multi-feature prognostic score studied for HER2+ eBC that includes clinical, pathological, and genomic information and was recently upgraded to also predict pCR likelihood in this specific subgroup of patients [14]. These methods essentially require pathologic tissue and invasive procedures that are available at diagnosis and after neoadjuvant treatment but are difficult to obtain during preoperative treatment to monitor the response. Multiple radiologic nomograms have been developed to anticipate the possibility of pCR and evaluate lymph node status after NACT; however, their use has not been suggested by international guidelines yet [15,16,17]. Circulating biomarkers may provide the missing clues in a non-invasive way. This review aims to focus on the available data regarding the potential utility of circulating biomarkers as prognostic and predictive biomarkers in the preoperative setting of eBC (Figure 1 and Figure 2).

## 2. Immune System and Tumor Cells: The Circulating Counterpart

### 2.1. Circulating Inflammatory Cells

Circulating inflammatory cells are involved in the process of metastasizing and seem to also have a role in the first phase of cancer promotion and progression, as observed from studies on metastatic BC patients [18,19,20]. Inflammatory cells, especially macrophages, drive tumor progression from the very initial phase, cooperating in cell dissemination in premalignant lesions [21]. These considerations suggest a potential role for these cells as biomarkers for eBC as well, and thanks to their feasible accessibility, lymphocyte counts are becoming interesting tools as surrogate markers for the outcome, immune response, and inflammatory status.

Pretreatment values of the neutrophil-to-lymphocyte ratio (NLR), platelet-to-lymphocyte ratio (PLR), and monocyte-to-lymphocyte ratio (MLR) may reveal predictions for treatment efficacy, pCR, and survival outcomes. A meta-analysis including more than 17,000 patients with early and advanced BC revealed an association between high pretreatment NLR and a poorer prognosis in eBC [22]. The identification of preoperatively high inflammatory indexes is predictive of a worse prognosis in the case of luminal eBC [23]. Especially in eBC, a high NLR may predict the necessity of more aggressive treatment to prevent a later progression or recurrence [24,25]. However, the data are conflicting. A retrospective study conducted by Grassadonia et al. suggested that low NLR may be an indicator of inadequate anticancer response and immunosuppressive background with potentially less efficacy of NACT [26].

Elevated platelet-related markers may also be associated with a poor prognosis in eBC. The platelet counts and platelet volume indexes seem to represent prognostic factors in eBC as indicators of an increased risk of metastasis [27]. In addition, an increased preoperative platelet distribution width-to-platelet count ratio (PDW/P) measurement represented an independent prognostic factor in patients with localized BC [28]. An elevated PLR has been found to adversely impact survival [29]. In a study conducted on 793 non-metastatic BC patients, an elevated preoperative PLR proved to be an independent prognostic marker for survival with a significant association with cancer-specific survival [30]. The other two studies showed that BC patients with a high PLR, before and after the initial treatment, respectively, had substantially lower survival rates than their counterparts [31,32].

Monocytes are particularly involved in the process of metastasizing, favoring cancer cell dissemination even in the very early stages, and managing the development of the metastatic niche [21]. The depletion of monocyte subtypes and macrophage counts may reduce early tumoral cell dissemination and slow cancer progression [33]. The lymphocyte-to-monocyte ratio (LMR) or MLR was shown to be significantly associated with a worse prognosis in eBC patients who underwent surgery after receiving standard neoadjuvant therapy [34].

The development of lymphopenia after NACT was not associated with pCR achievement in a retrospective analysis regarding exclusively TN eBC [35]. However, the same analysis showed a correlation between a lower absolute circulating monocyte count after NACT and pCR. In the Peng et al. analyses, a low LMR reflects a greater capacity to respond to neoadjuvant treatments, resulting in the only independent predictive factor for the efficacy of NACT [36]. Similar considerations have been proposed about PLR and NLR [37]. The prognostic role of these biomarkers has been demonstrated in all BC subtypes, particularly in patients with HER-2 positive eBC who did not receive trastuzumab, suggesting a role in the modulation of the inflammatory response of trastuzumab [38]. Regarding pathological response after NACT, a low pretreatment NLR has been associated with a higher rate of pCR [39]. However, a low NLR was independently predictive for febrile neutropenia, as shown in another study [29].

Additionally, the preoperative systemic immune-inflammation index (SII: calculated by multiplying the neutrophil count by the platelet count and dividing the result by the lymphocyte count) showed to potentially give prognostic information before surgery [40]. A high SII showed a correlation with a worse pCR rate in triple-negative Asian eBC patients [41]. Furthermore, patients with low SII showed better OS and DFS in two retrospective studies [42,43].

Certainly, the retrospective nature of the majority of the data available and the retrospective findings of ongoing studies (NCT05468710) will be a clear limitation for the clinical utility of inflammatory indexes in terms of prognostic validity. Prospective studies are awaited to thicken the body of evidence about the prognostic value of immune circulating cells in eBC.

### 2.2. Circulating Tumor Cells

CTCs are cancer cells that are detectable in the bloodstream [44]. CTCs have been studied in the metastatic setting to follow the course of BC disease and are used to predict survival or response to treatment [45,46]. In metastatic disease, it is possible to detect at least 1–5 CTC/7.5 mL [47]. Despite the rarity of these cells, it is possible to determine CTCs in peripheral blood for stage I–III eBC thanks to new detection methods, suggesting that these cells may guide the occult dissemination even if there is no lymphatic involvement [48,49].

Detection techniques have evolved over the years. With the current techniques, CTSs are initially isolated, followed by a phase of enrichment, and then counted [50]. CTCs are mixed and hidden with the billions of other blood cells in circulation. The isolation phase is crucial, but the enrichment phase is more important. This phase may be achieved by positive or negative immunoselection methods using different antibodies directed at membrane antigens such as the epithelial cell adhesion molecules (EpCAM), as used by the immunomagnetic enrichment tool CellSearch^®^ [51]. Immune-related methods can combine different antibodies, targeting surface proteins and extracellular matrix components, to capture CTCs that normally do not express EpCAM (as observed after epithelial-to-mesenchymal transition) [52]. CTCs may be quite heterogeneous, and for this reason, other methods of enrichment, affording physical properties, have been developed [53]. Among antigen-independent technologies, we enumerate size-dependent methods or techniques evaluating deformability and electrical (conductive and dielectric) characteristics that differ between CTCs and blood cells [54]. Although there are hundreds of quantitative techniques used to detect CTCs, only a few have been approved or have gained the clinical acceptance in BC that CellSearch^®^ has. CellSearch^®^ has been the most used method in trials, with some detection limits determined by the fact that this method does not identify EpCAM-negative cells [43].

The quantification of CTCs may be an effective non-invasive tool to predict disease aggressiveness and monitor treatment response in the preoperative setting. Many studies have been conducted using CellSerch^®^ to detect the CTC count in the bloodstream before and after NACT in the eBC, showing a correlation between the CTC count and survival outcome (Table 1). In general, the median detection rate observed in the studies is around 22%, except for high T-stage tumors or inflammatory BC, which present a higher quantity of CTCs [55,56]. In Bidard and colleagues’ meta-analysis involving more than 2000 eBC patients, the detection of more than one CTC before systemic treatment predicts a worse survival outcome and a higher risk for early recurrence, suggesting the utility of intensifying treatment [57]. It highlighted the correlation between tumor dimension and CTCs, particularly for high T-stage tumors such as T4d, or inflammatory breast cancer, and HR-eBC [57]. CTCs remain as independent prognostic factors, as well as excluding T4d tumors from the final analysis. In each study reported in Table 1, a correlation between CTC count and pCR was not observed. However, in Bidard and colleagues’ meta-analysis, HR+ eBC patients with no CTCs detected before NACT presented a moderate increase in achievement of pCR rather than patients with one or more CTCs at basal evaluation (*p* < 0.01) [57]. In addition, the substudy of the NeoALLTO trial showed a correlation between lower pCR rates and patients with detectable CTCs at any time point or before surgery without statistical significance [58].

Detecting CTCs in eBC is not easy because CTCs are not present in all patients [59], but the quantification of CTCs (when identified) remains an independent prognostic biomarker in preoperative settings, predicting worse survival outcomes and leading to patient stratification. There is still an open debate on whether CTC’s quantitative information can be used for escalating or de-escalating treatments in the neoadjuvant approach.

The qualitative characterization of the molecular features of CTCs could also contribute to complete prognostic information. Different methods have been developed to obtain phenotype and molecular information. The quantitative reverse transcription PCR (RT-PCR) is a potential method, and different kits for detection are using these techniques, such as the AdnaTest Breast Cancer Select/Detect (AdnaGen AG, Langenhagen, Germany) or CircleGen CTC RT-qDx (Syantra, Calgary, Canada) assays [70,71]. These techniques foresee the immunomagnetic separation of CTCs, then the cells undergo a lysis process for RNA isolation and subsequent cDNA synthesis for the analysis of CTC-related mRNAs to relative tumor genes by RT-qPCR [72]. This method has been used by Wang et al. to analyze the presence of CTC in 221 eBC patients’ blood samples before and after treatment (chemotherapy and surgery), with a detection range of 15% to 45% [72]. In the study, 69.2% of patients had a positive signal in at least one cancer-associated marker gene in the analyzed CTCs, with a moderate concordance to the biological subtype of the diagnosed tumor. Furthermore, in the study, the expression of different markers during the course of the treatment was evaluated, showing an increased expression of proliferative markers from baseline to cycle 8 and a concomitant reduction of HER-2 and epithelial markers [72]. However, the discordance between CTCs has been widely explored and represents both one of the hallmarks of CTCs and their limits at the same time [73]. In a subanalysis of the GeparQuattro Trial, HER-2 overexpressing CTCs were also observed in patients with HER-2 negative eBC [62]. CTCs were detected by analyzing the HER-2 transcript with RT-PCR, especially in patients with high-stage disease, as a marker of aggressiveness [62]. It was hypothesized to observe HER-2 expression to monitor anti-HER-2 treatment efficacy. In particular, in the preoperative setting for HER-2+ebc, it is very important to predict in the most precise way the possibility of pCR to choose an intensified strategy (with pertuzumab-trastuzumab and sequential chemotherapy anthracycline-based) even with the risk of no pCR and the necessity of further intensive treatment with T-DM1 [10,11].

The arrival of checkpoint inhibitors in the treatment algorithm for BC has highlighted the importance of PD-L1 expression, particularly in the metastatic setting. After the results of the Keynote 522 study, pembrolizumab is indicated in high-risk early TNBC (independently of PD-L1 status) combined with paclitaxel/carboplatin and followed by anthracycline/cyclophosphamide, guaranteeing a major pCR achievement and survival outcome [12]. However, it is not very clear how to individuate those patients who could achieve greater benefit from this, and currently, there is no univocal strategy for adjuvant treatment where there is a lack of pCR [74]. PD-L1 can be assessed on CTCs, and it may provide more information for response to treatment [75].

Tools for stratifying patients and defining the real benefit of escalated treatments are needed to reduce unnecessary overtreatment and maximize the therapeutic effect. However, it is common to observe different transitional phenotypes and molecular heterogeneity derived from the epithelial to mesenchymal transition of CTCs in the bloodstream, limiting their applicability in clinical practice [76]. The molecular heterogeneity of CTCs may be more representative of the entire clones of metastatic BC, which is notoriously a heterogenous disease but is also present in the earlier stages [77]. Rossi et al. analyzed the presence of CTCs at different time points after surgery in a limited case series of eBC, observing the presence of CTCs in 90.9% of such cases and a progressive clearance of them after adjuvant treatment and six months after surgery [78]. In the study, even with the persistence of CTCs six months after surgery (*n* = 7 patients), none of the patients relapsed during the follow-up, and the authors consider that these cells may have a low metastatic potential. Molecular analysis NGS based on single CTCs revealed different copy number aberration (CNA) at baseline compared to the subsequent different time points, highlighting the importance of timing of surgery in CTCs selection and limitation of CTCs reservoir [78]. Moreover, the analysis showed altered pathways involved in the mechanism of the metastasis process (such as type I interferon-associated genes) in CTCs stored at diagnosis [78]. The observation of the CNA burden, compatible with the metastasis process and progression signals from the earliest stages, may suggest a prognostic relevance of molecular CTC information in predicting a major risk of recurrence and the need for more intensive treatments.

Of note, there is a close relationship between proinflammatory markers and the presence of CTCs in peripheral blood [79]. In particular, this connection has been observed in the metastatic setting [19], but it has also been recently demonstrated with eBC [80]. Some studies described a great interplay between proinflammatory indices and CTC subtypes in eBC patients [80,81,82]. In particular, epithelial-CTCs were closely related to lymphocytopenia, a low monocyte count, and high NLR and PLR [80]. Conversely, the presence of epithelial-to-mesenchymal transition phenotype CTCs was significantly correlated with an elevated MLR [80]. The combination of inflammation-based scores and CTCs may improve the prognostic stratification of patients [82].

The key question is if we are ready to use CTCs in daily clinical practice [83]. Among the limits of the different types of techniques (such as different isolation methods or the different amounts of blood in sample collections) are heterogeneity and a scarce detection rate in the earliest settings; currently, CTCs are unable to give the information necessary to establish clinical utility [84]. Recent guidelines recommend not using CTCs to guide clinical decisions because strong data are not available yet [2]. Nevertheless, recent studies are ongoing to explore the potential implementation of screening techniques and risk evaluation before choosing a therapeutic strategy in the preoperative setting (Table 2) [85].

## 3. Circulating Cell-Free Nucleic Acids

### 3.1. Circulating Tumor DNA

CtDNA detection from a liquid biopsy is becoming an interesting biomarker for eBC research in diagnosis, classification, prognostication, and treatment selection [86]. Tumoral cells release DNA fragments in the bloodstream after various mechanisms of cell death (necrosis, apoptosis, autophagy, necroptosis, or ferroptosis) or active secretion that contribute to enriching the amount of cfDNA [87]. In cancer patients, ctDNA represents a very small fraction of the entire amount of cfDNA, and variations depend on the stage, type of cancer, localization, and vascularization [88].

The techniques developed for ctDNA extraction and quantification are progressively evolving and becoming more sensitive, even in cases of low amounts of disease (such as eBC) [89]. Tumoral-derived DNA fragments present genomic alteration as point mutations, deletions, copy number variations, rearrangements, and chromosomal variations [90]. The possibility of studying ctDNA variation ranges from single-gene methods (with PCR-based methods, such as droplet digital polymerase chain reaction, ddPCR) to the possibility of a wide analysis with NGS methods and whole genome/exome sequencing approaches (increasing the number of variations analyzable with a multigene panel analysis [91,92].

The use of ctDNA analysis is increasing in the preoperative setting. CtDNA first presents clinical utility in treatment selection for advanced BC by characterizing the response to target or resistance to treatment [93,94]. In early diagnosis, the objective is to detect early-stage cancer or anticipate diagnosis in asymptomatic subjects but also to monitor the treatment response [95]. The sensitivity of ctDNA analysis depends on both the amount of blood in the samples and the number of mutations screened [89]. Mutational detection in ctDNA has been observed years before tumor diagnosis [96]. Liquid biopsy may become a non-invasive method for diagnosis when combined with classical radiological screening methods, increasing the adherence of patients to screening programs. However, there are limits regarding the low detection rates of tumor fractions of cfDNA in the bloodstream in cases of low-burden breast tumors or ductal carcinoma in situ [4].

Currently, there is no data about the clinical validity and utility of ctDNA in eBC screening; results from prospective trials are awaited (Table 3). Among them, the NCT03085888 trial, a large prospective observational study enrolling a wide population of approximately 100,000 women undergoing screening procedures for BC, aims to develop an early detection assay using high-intensity sequencing of circulating cell-free DNA (based on broad genomic coverage and deep sequencing) [97]. In the perioperative setting, the possibility of earlier detection of the disease with ctDNA may be useful to define the proper timing of surgery and anticipate it, particularly in cases of rapid disease growth with new detectable ctDNA variants due to multiplications of cellular clones [89]. The majority of data about the use of ctDNA comes from studies about the monitoring of residual disease (MRD) after curative therapy, identifying patients with a high risk of relapse [98,99]. The first prospective study for the detection of ctDNA in eTNBC after active curative treatment has been recently published, assessing its clinical utility in guiding therapy and speeding up the disclosure of recurrent metastatic disease [100].

In general, it has been demonstrated that ctDNA detection in cancer patients in the neoadjuvant treatment phase is associated with worse outcomes [101]. In the case of eBC treated with NACT, the treatment response may be monitored with a longitudinal quantitative analysis of ctDNA. The clearance of ctDNA during neoadjuvant treatment has been shown to be informative about prognosis and potentially may help to adjust the course of the treatment as a meaningful surrogate of outcome [102]. In the majority of cases, studies contemplate primary tumor sequencing to distinguish somatic genetic variants that can be detected in plasma to monitor the disease [103]. CtDNA may provide useful prognostic information and fill in the gaps regarding the reasons why patients recur despite achieving pCR after NACT and, conversely, the possibility of not recurring even with the presence of residual disease after NACT [104,105,106]. The meta-analysis of Papakonstantinou and colleagues sums up all the evidence regarding the prognostic role of ctDNA at baseline and after NACT as a biomarker for a worse risk of relapse in eBC patients [107]. Most studies included in the meta-analysis enrolled exclusively (or for the majority) TNBC patients (the most common type of BC receiving NACT) [75,106,108,109,110,111,112]. The triple-negative biology is notably the most aggressive, and triple-negative tumors are more likely capable of releasing larger quantities of ctDNA; this may have driven the final results of the meta-analysis [99,105]. In the meta-analysis, it was highlighted the capacity of ctDNA detection to predict OS (HR 19.1, HR 4.00 before and after NACT, respectively) and risk of relapse (HR 4.2, HR 5.67 before and after NACT, respectively) [107]. Unexpectedly, in the same analysis, ctDNA did not correlate to pCR. The sole trial included in the analysis by Papakonstantinou et al. describing a correlation between ctDNA levels and pCR was a subanalysis of the result of the NeoALLTTo trial [113]. In the NeoALLTO trial, ctDNA detection before NACT combined with anti-HER2 treatments resulted in decreased pCR rates; conversely, HER2+ eBC patients with no detectable ctDNA before treatment achieved much more pCR [113]. A possible limitation of the studies may derive from the heterogeneity of criteria for ctDNA quantification with different target mutations (single nucleotide variants, TP53, PIK3CA, mTOR, and AKT) and different diagnostic procedures (NGS, ddPCR) [114]. Furthermore, clonal evolution in the initial stages of the disease is less represented compared to advanced disease, and the tumor mutational burden might be less informative [89].

Results are awaited, and many studies are ongoing regarding the ability of ctDNA to anticipate pCR achievement (Table 3). Prospects are good, but CtDNA-detection-guided escalation and de-escalation prospective studies are needed to affirm the real clinical utility of ctDNA as treatment guidance in the neoadjuvant phase.

### 3.2. Fragmentomics

The study of fragmentation patterns of cfDNA, also known as “fragmentomics’’, has enlarged the potentiality of liquid biopsy in cancer diagnosis [115]. Unlike the ctDNA analysis, where the mutational status characterization is essential for its quantification, the epigenetic analysis does not need such determination [116,117]. This appears to be helpful in early disease where the mutational burden is not so expressed due to low clonal tumor development [89]. Fragmentomics analysis is gaining success as a non-invasive, cost-effective new method in BC [5].

CfDNA fragments are released in the bloodstream after death processes, with different lengths depending on the cell’s tissue of origin and the secretion method [118]. It has been widely debated whether fragments of ctDNA are longer or shorter than cfDNA derived from healthy cells [119,120]. Most studies involving BC patients assume that the ratio between shorter and longer fragments, defined as the cfDNA integrity index (cfDI), may give information about the totality of ctDNA fragments [5].

In previous work from our group, we provided an overview of all available data regarding the use of cfDI as a diagnostic biomarker in BC [5]. In the majority of studies, the ratio between long and short fragments correlated with the tumor presence and tumor stage [121,122]. A high cfDI appeared to be an independent prognostic factor for poor survival outcomes (worse OS and higher risk of relapse) in BC patients [121,123,124]. The trend describing the variation in fragment quantity and cfDI showed contrasting results in different studies [5]. Wang et al. observed an increasing trend associated with tumor shrinkage (*p* < 0.05) in the course of NACT and a higher cfDI in eBC patients exposed to NACT who achieved pCR at surgery [125]. Conversely, two other studies showed a progressive clearance of cfDNA and cfDI in the course of NACT [126,127]. A downtrend of short and longer ALU amplicons from cycle one to six of NACT was observed in pCR patients, whereas an increase was observed in non-responders (*p* = 0.033) [127]. The benefits from fragment size analysis are still hindered by many controversial results determined by different methods of pre-analytical analysis, different populations, and discrepancies among authors that complicate data comparisons [5,120].

Fragment characterization can also provide information about other epigenetic features, such as nucleosomal footprints and end-fragment signatures that distinguish ctDNA [115]. CfDNA is composed of nucleosome-protected DNA fragments released into the bloodstream [128]. The analysis of plasma DNA with whole-genome sequencing could be useful in identifying transcription start sites (TSSs) and the different nucleosome occupancy that characterizes tumoral gene signatures associated with different gene expressions or silencing [129]. Nucleosome position may be helpful to identify the tissue of origin and to distinguish cancer from benign tumors, improving early diagnosis [130,131].

The study of post-translational modifications (PTMs: acetylation, methylation, phosphorylation, ubiquitination, and sumoylation) of nucleosomes may add useful information to eBC diagnosis. Different PTMs have been identified in different cancer histologies and correlate in many cases with prognosis-determining transcriptional activation or repression [132]. In BC, different PTMs of histones also correlate with the different molecular subtypes (e.g., luminal, basal-like), and the quantification of nucleosome PTMs may distinguish ductal carcinoma in situ from invasive eBC [133,134]. Exploring the epigenomic profile of cfDNA can also be used to predict the efficacy of NACT from the first cycles of therapy. Mapping nucleosome positioning in patients with eBC receiving NACTs with different responses allowed Yang et al. to observe that eBC patients presented distinct nucleosome footprints at TSSs compared to healthy donors, with differences according to the pathological response [135]. In eBC patients undergoing NACT, it has been observed that high levels of circulating nucleosomes before treatment correlate to worse survival outcomes [136]. The measurement of absolute levels of circulating nucleosomes and the description of PTMs provide an exciting new avenue for new non-invasive diagnostic methods and new tools for cancer monitoring under treatment in a preoperative setting. However, the clinical utility of the cfDNA nucleosome footprint has not been fully confirmed yet, and further results and studies are awaited (such as NCT03992521).

The characteristics of the ends of circulating DNA fragments are also of interest. CtDNA preferred ends are different from those of cfDNA released by normal cells because they correspond to different loci and are more easily susceptible to the action of deoxyribonucleases (DNases) according to nucleosome positioning and chromatin winding [5]. End signature analysis with WGS techniques could have good sensitivity and specificity for cancer detection, representing a cost-effective way to detect eBC thanks to the wide presence of preferred end coordinates across the genome [137]. Moreover, different DNases are expressed in various tissues and cancers, determining specific end motifs [138]. This peculiarity may be exploitable for the early diagnosis of various cancers [139]. Epigenomic features are predominantly studied in preclinical research; further evidence is awaited to determine the effective utility in eBC diagnosis and monitoring.

### 3.3. DNA Methylation Signature

Methylation analysis can identify the different origins of cfDNA; the majority of it is released by hematopoietic cells, and ctDNA presents different methylation features that can be detected [89]. The alternation of hypermethylation and hypomethylation in tumoral DNA allows the silencing of tumor-suppressor genes and the activation of oncogenes, respectively, promoting cancer progression [140,141,142]. The methylation signature differs among various tissue and various cancers; therefore, the analysis of it may improve cancer diagnosis, in particular for difficult diagnosis [143]. This detection method may also be useful for tumor diagnosis when a low amount of ctDNA is released in the bloodstream and scarce DNA variations are detectable (as eBC with low cancer burden and low clonality) [144]. Liu et al. explored the potential clinical utility of cfDNA methylation as a biomarker for eBC diagnosis in 203 female patients with suspicious breast lesions [144]. They showed that combining methylation features from liquid biopsy with standard imaging procedures (ultrasonography and mammography) was particularly helpful for Breast Imaging Reporting and Data System (BI-RADS) in four patients that usually present high rates of false-positive and unnecessary invasive biopsies. The NCT04822792 trial is currently ongoing and enrolling healthy female subjects who are attending gynecological and mammographic control to test a cfDNA methylation-based model for detecting eBC. This non-invasive method has shown its utility in anticipating recurrence, detecting MRD, and evaluating adjuvant treatment efficacy/resistance [145,146,147].

In the preoperative setting, Moss and colleagues used a breast-specific DNA methylation signature to detect ctDNA in localized BC before neoadjuvant treatment [147]. The study, conducted on 235 patients, distinguished aggressive molecular cancer profiles with high ctDNA levels. In these patients, NACT resulted in a dramatic decrease in ctDNA levels, and the persistence of ctDNA after NACT reflected the existence of residual disease at pathology analysis [148]. In a study involving 83 women with localized advanced BC, DNA methylation changes were associated with response to NACT and survival [149]. Chemotherapy proved it could vary methylation signatures, determining hypomethylation at CpG islands and hypermethylation in non-CpG islands, and affect drug response through gene silencing or activation [150]. The effect of NACT on BC cells was also demonstrated by Luo et al., showing an incremented hypomethylation of quiescence genes that may become therapeutic targets [151,152]. The epigenetic signature of specific genes could also be analyzed to predict pCR, as suggested by Pineda and colleagues who observed the role of methylation of the FERD3L and TRIP10 genes in TNBC [153]. Large-scale validation studies are awaited to use DNA methylation biomarkers in clinical practice [154].

### 3.4. Non-Coding RNAs: miRNA and circRNA

The majority of the human genome can be transcribed; however, the protein-coding DNA accounts for only 2% of the entire genome sequence [155]. Most of the sequence is transcribed into noncoding RNA (ncRNA), which does not evolve into protein synthesis [156]. NcRNA is constituted of functionally important types of ncRNAs, including transfer RNAs (tRNAs), ribosomal RNAs (rRNAs), and small RNAs such as microRNAs (miRNAs) and circular RNAs (circRNA), among many others [157]. The circRNA–miRNA–mRNA axis is related to the mechanisms of tumor progression and development in BC [158].

MicroRNAs are single-stranded RNAs (20–25 nucleotides) that play a role in regulating mRNA translation [159]. In BC, they act as oncogenes (oncomiRNAs) or suppressors and present an important role in cancer progression [160]. MicroRNAs can be detected in tissue or blood samples where they are contained in exosomes or extracellular vesicles, but they can also be free in the plasma [161,162]. Recent studies have shown that some miRNAs are cancer-specific and may be used as diagnostic tools [163,164,165]. In BC, miR-21 resulted in being more specific than classical serum markers (i.e., Ca 15.3) and was useful in clarifying BC diagnosis and monitoring [166]. Moreover, the histologic subtype strongly influences the type of miRNA expression in blood or tissue [167,168]. A multi-biomarker tool combining miRNA and ctDNA is under evaluation to obtain a non-invasive, affordable diagnostic method (NCT04906330, Onco-liq).

CircRNA instead presents a circular configuration that gives it major stability in the bloodstream [158]. CirCRNA act as miRNA expression regulators; they catch miRNAs similar to a sponge and are responsible for their activity on genomic expression while promoting cancer activity [159]. Some circRNAs are upregulated in BC and may be used as biomarkers to anticipate diagnosis thanks to their more stable conformation [169,170]. The analyses conducted by Sarkar and Diermier identified circ0001785 as a diagnostic biomarker for BC and in correlation with disease stage and histologic subtypes [171]. They found that circ0001785 levels were lower after surgery in eBC and that its detection may efficiently recognize disease relapse.

NcRNAs are under evaluation to understand their potential in monitoring the NACT response of eBC patients. Shobani et al. provide a complete review about ncRNA in BC, highlighting multiple ncRNA connected to response to NACT [172]. The fluctuation of miRNA levels during NACT has been studied to evaluate a relationship with response to therapy and a potential prognostic role [173,174]. A recent meta-analysis revealed that 60 miRNAs are related to NACT response [175]. In particular, a high baseline miR-7 level was associated with a higher pCR rate (*p* = 0.0004). The same analysis indicated 26 different miRNAs related to survival outcome, among which miR-21 was associated with a poor prognosis if high levels before and after treatment were detected [175]. The utility of miRNA in defining potential responders to NACT could also be useful for eBC patients with histologic subtypes that often do not strikingly respond to NACT. Zhang et al. identified specific circulating miRNAs (miR-718, miR-4516, miR-210, and miR-125b-5p) as predictive markers for NACT response and prognosis in HR+/HER-2-patients, whereas miR-222 was observed in the HR+/HER-2+ cohort [176]. Circulating miRNAs have also been studied in a substudy of the NeoALLTO trial in patients treated with lapatinib + trastuzumab before and after two weeks. Fifty-two different circulating miRNAs were able to predict pCR achievement, hypothesizing their role as biomarkers in de-escalating treatment in patients with a molecularly favorable phenotype [177]. A similar analysis has been conducted on the patients enrolled in the Geparsixto trial (MiR-155 and miR-301) [178].

Finally, miRNA might also be responsible for resistance to treatments, and identifying resistance biomarkers is particularly useful to individualize treatment and avoid toxic therapies. Some miRNAs have a regulatory behavior controlling intracellular signaling that is translated into higher drug efflux or DNA repair and cell cycle variations, resulting in drug resistance [179,180]. In their review, Zangouei et al. provide a list of all known miRNAs that may determine sensitivity or resistance to anthracyclines at various steps in the cellular machine [179]. MiRNAs have also been investigated as biomarkers for cardiac toxicity in eBC patients undergoing NACT. Levels of let-7f, miR-19a, miR-20a, miR-126, and miR-210 were lower in eBC patients who experienced cardiac side effects after NACT [181]. Zhang M et al. defined the known circRNA-miRNA as being associated with increased resistance to anthracyclines or taxane [158]. Studies are ongoing to test if changes in miRNA expression can be used to indicate drug failure in eBC patients (NCT04771871). Defining patients with multi-drug resistance or potential toxicities to anthracyclines would contribute to identifying the best treatment choice and supporting chemotherapy de-escalation, enabling clinicians to safely withhold anthracyclines [182].

More and more ncRNA have been cataloged and connected to the prediction of treatment response in eBC [183,184,185]. However, low sensitivity and low specificity of detection techniques limit miRNA analysis, and new methods are under development. Li and colleagues, for instance, devised a new graphene oxide (GO)-based qRT-PCR method for detecting miRNAs and distinguishing NACT responders in eBC patients [186]. Further results are needed to better understand how miRNA may become useful for patient stratification and for defining personalized treatment in eBC.

## 4. Methods

Our review was performed by following the PRISMA guidelines for reporting systematic reviews and meta-analyses (Figure 3) [187]. We conducted a review of English-language literature until November 2022. Our research was performed using the web databases Medline and PubMed. The aim of our research was to find the relevant studies dealing with circulating biomarkers in eBC patients undergoing pre- treatment. We used the following search terms: biomarkers OR liquid biopsy OR ctDNA OR circulating tumor cells OR circulating inflammatory cells OR fragmentomics OR ncRNA OR miRNA OR methylation AND early OR pre-operative AND breast AND cancer OR tumor. After reading the abstracts, we more thoroughly analyzed the article’s full text. We also checked through the references of each article in order to identify further interesting studies. In particular for the creation of Table 1 we performed an English language literature research until November 2022 using the following search terms: circulating inflammatory cells OR CTC AND early OR pre-operative AND breast AND cancer OR tumor. For the construction of Table 2 and Table 3, we queried the ClinicalTrials.gov database (Available online: https://clinicaltrials.gov/ (accessed on 20 November 2022)) using the following search terms: breast cancer AND liquid biopsy OR ctDNA OR CTC. From the totality of the trials resulting from the research, we selected only the studies involving eBC patients in the pre-operative setting that are currently ongoing (Figure 3).

The search process according to the PRISMA Guidelines. * 20 studies considered for Table 1, 12 active trials considered for Table 2, 8 active trials considered for Table 3. 

## 5. Conclusions

The intriguing results of novel circulating biomarkers will be useful in evaluating more precisely the prognosis of eBC; however, their use has not been established yet in clinical daily practice [2]. Further studies are needed to reduce the current limitations regarding the analysis. In particular, pre-analytical variables may alter the sample processing, and various testing procedures may provide different results that are non-comparable with each other, lacking analytical validity [5,18,94]. Moreover, clinical validity has not been completely demonstrated yet due to the insufficient evidence of most assays [98].

Currently, not all the assays are equal to the others. Too many differences between the detection techniques do not allow for a defined, univocal method and consequently, we are far from a definitive conclusion regarding which test has the most predictive value for the clinical outcome. Accordingly, the clinical utility for cancer screening, response monitoring, and evaluation of treatment efficacy of circulating biomarkers is not completely defined and not applicable except in clinical trials. The gap between bench and bedside can be filled when circulating biomarkers are used regularly in wide prospective trials on neoadjuvant approaches in eBC, establishing their direct role in the decisional process. Results of ongoing trials are awaited to improve tools for escalating and de-escalating approaches in eBC. The combination of clinical, pathological, genomic, laboratory, and radiologic findings will contribute to a more precise stratification of the patients and the diseases. The development of predictive models and nomograms with the combination of multiple features should be the future challenge to reduce the need for invasive approaches.

## Figures and Tables

**Figure 1 cancers-15-00096-f001:**
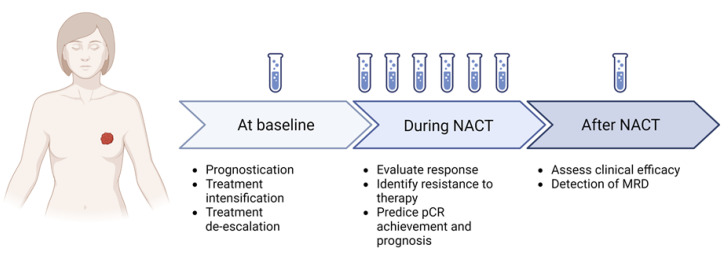
Potential application of liquid biopsy in eBC. Created with Biorender. Available online: https://app.biorender.com/ (accessed on 15 November 2022).

**Figure 2 cancers-15-00096-f002:**
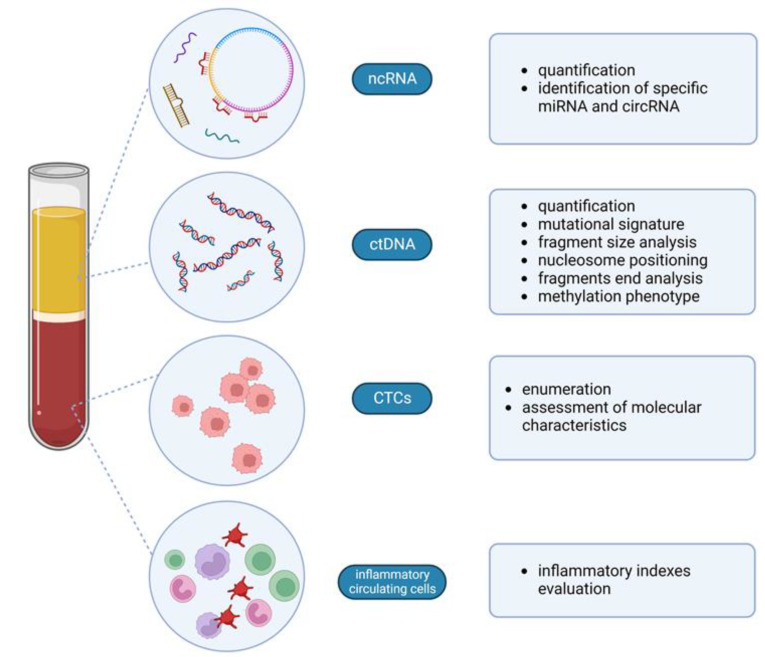
Various circulating biomarkers are useful in the preoperative phase in eBC. Created with Biorender. Available online: https://app.biorender.com/ (accessed on 15 November 2022).

**Figure 3 cancers-15-00096-f003:**
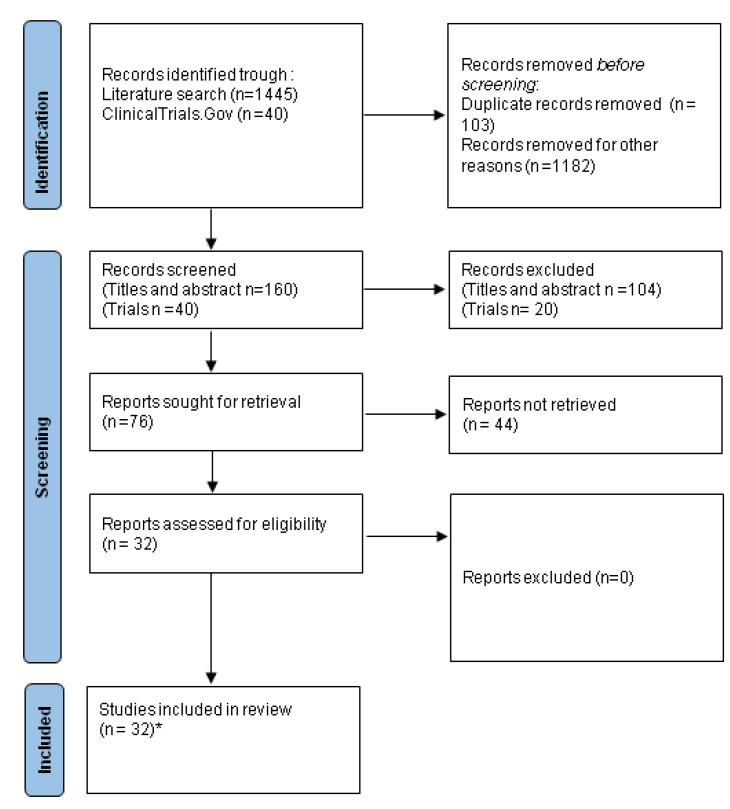
The search process according to the PRISMA Guidelines.

**Table 1 cancers-15-00096-t001:** Studies evaluating a correlation between CTCs and a worse prognosis in eBC in case of CTC detection.

	CTCs Detection Rate before NACT	CTCs Detection Rate after NACT	Results
IMENEO [57,60]	25%	17%	OS (*p* < 0.001; HR 3.93, 2.00–5.45), DDFS (*p* < 0.001; HR 3.73, 2.82–4.90), LRFI (*p* < 0.001, HR 3.02; 1.88–4.75))
GeparQuattro [61,62]	22%	10%	DMFS (*p* < 0.001; HR, 3.72, 1.89–7.32), OS 0 (*p* < 0.01; HR 4.54, 1.97–10.49)
REMAGUS 02 [63]	23%	17%	RFS *p* = 0.013 (HR N/A)
NEOALTTO [58]	11%	N/A	No prognostic results
NEOZOTAC [64]	18%	N/A	No prognostic results
MD Anderson [65]	N/A	27%	RFS (*p* = 0.03; HR 5.25, 1.34–20.56), OS (*p* = 0.03; HR 7.04, 1.26–39.35)
MD Anderson [55]	54%	N/A	PFS (*p* = 0.02; HR 0.60, 0.37–0.98), OS (*p* = 0.03; HR 0.59, 0.35–1.00) *
MD Anderson [66]	24%	N/A	PFS (*p* = 0.005; HR 4.62, 1.79–11.9), OS (*p* = 0.01; HR 4.04, 1.28–12.8)
BEVERLY-2 [67]	35%	7%	DFS (*p* = 0.01; HR 3.69, 1.34–10.21)
BEVERLY-1/-2 [56]	39%	9%	DFS (*p* < 0.01; HR 2.80, 1.65–4.76) OS (*p* < 0.01; HR N/A)
Serrano et al. [68]	70.08% (10 mL)	54.1% (10 mL)	OS (*p* = 0023; HR N/A)
Janni et al. [69]	20.2%	N/A	OS (*p* < 0.001; HR 1.44, 1.81–3.29), DFS (*p* < 0.001; HR 2.08, 1.69–2.56), DDFS(*p* < 0.001; HR 2.20, 1.74–2.78), BCSS (*p* < 0.001; HR 2.54, 1.910–3.38)

Abbreviations: BCSS = breast cancer-specific survival; DFS = disease free survival; DDFS = distant disease free survival; DMFS = distant metastasis free survival; HR = hazard ratio; LRFI = locoregional relapse free interval; N/A = not available; OS = overall survival; PFS = progression free survival; RFS = recurrence free survival. * Patients with fewer than five CTCs had a better PFS and OS than patients with more than five CTCs detected.

**Table 2 cancers-15-00096-t002:** Ongoing trial about CTCs in the preoperative setting.

Trial	Patients (n)	Objective	Status
NCT04239105	N/A	To develop a eBC screening test. To evaluate the efficacy of NACT and prognosis.	Not yet recruiting
NCT03511859	210	To develop a eBC screening test.	Unknown, not yet posted
NCT01322750	3125	To develop simple, reliable, cost-effective, and clinically relevant breast cancer screening test.	Recruiting
NCT03842176	90	To monitor response during neo/adjuvant treatment.	Unknown
NCT03709134	100	To investigate the role of CTCs predicting response to NACT.	Recruiting
NCT05326295	1000	To predict treatment response of NACT, surgery and adjuvant chemotherapy. To evaluate the prognostic role of CTCs.	Recruiting
NCT04993014	80	To predict treatment response of NACT, surgery and adjuvant chemotherapy. To evaluate the prognostic role of CTCs.	Recruiting
NCT04059003	200	To evaluate changes of CTCs and the efficacy of NACT for TNBC.	Recruiting

Abbreviations: CTCs = circulating tumor cells; eBC = early breast cancer; NACT = neoadjuvant chemotherapy; TNBC = triple negative breast cancer.

**Table 3 cancers-15-00096-t003:** Ongoing trials regarding the use of ctDNA for early cancer detection and monitoring of the NACT response, including breast cancer patients.

Trial	Participant	Patients Characteristics	Biological Rationale	Endpoints	Status
NCT03881384	200	eBC	ctDNA clearance level during NACT and detection of MRD after surgery	ctDNA detection and clearance during NACT	Enrolling
NCT04276337	50	HER-2+ stage III eBC	ctDNA monitoring during during NACT (TCHP Regimen)	pCR rate	Active, Not Recruiting
NCT05050890	38	eBC	ctDNA monitoring during during NACT	ctDNA detection and clearance during NACT	Active, Not Recruiting
NCT02546232	196	I-IV stage BC	ctDNA analysis and molecular characterization for for the Optimal Selection of Treatment Regimens (NACT or treatments for aBC)	Correlate molecular changes to pathological response	Unknown
NCT03709134	100	eBC	role of CTCs and ctDNA in predicting response to NAC,	pCR rate	Recruiting
NCT04223492	100	eBC	Combination of standard screening techniques to liquid biopsy (CTCs, ctDNA)	pCR rate	Unknown
NCT03973034	300	healthy subjects, benign breast tumors, eBC	ctDNA test model for early screening of breast cancer	Diagnosis rate	Unknown
NCT03085888	99,481	healthy subjects involved in mammogram screening	Detection of breast and other invasive cancers analyzing cfDNA	Performance of the detection test	Active, Not Recruiting
NCT04241796	6662	healthy subjects	cfDNA and machine learning to detect a common cancer signal across >50 cancer types	Performance of Multi-Cancer Early Detection Test	Completed
NCT04972201	2305	healthy subjects	cfDNA mutation, miRNA, DNA methylation assays to detect cancer	Sensitivity for cancer detection and tissue of origin of the assays	Recruiting
NCT05227261	1643	healthy subjects	Anticipation of cancer diagnosis	Positive predictive value, Negative predictive value of the blood ctDNA test in early detecting cancers	Recruiting
NCT05235009	500	healthy subjects	Multi-cancer early detection	Sensitivity and specificity of the test	Recruiting

Abbreviations: aBC = advanced breast cancer; BC = breast cancer; cfDNA = cell free DNA; ctDNA = circulating tumor DNA; CTCs = circulating tumor cells; eBC = early breast cancer; miRNA = microRNA; MRD = minimal residual disease; NACT = neoadjuvant chemotherapy; TCHP = Docetaxel + Ciclofosfamide + Herceptin + Pertuzumab; pCR = pathological complete response.

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
