# Peer review of "Potential Impact of Preoperative Circulating Biomarkers on Individual Escalating/de-Escalating Strategies in Early Breast Cancer"

_cancers, 2022, doi:10.3390/cancers15010096_

Round 1

Reviewer 1 Report

Potential Impact of Preoperative Circulating Biomarkers on Individual 2 Escalating/de-Escalating Strategies in Early Breast Cancer. The authors provided a review of biomarkers on escalating and de-escalating strategies in early breast cancer. The article is comprehensive and novel. The following revisions should be made.

·       1. On lines 56 and 57 on page 2, the authors should use a similar protocol for referencing.

·       2. Section 2, 3, and 4 contains circulating inflammatory cells and tumor cells, etc. It is better to rename this section as background and make the aforementioned sections as subsections.

·       3. At line 117 on page 4, it seems to be PLR, not PRL.

·       4. There is a need for a section and a table to provide the comparison of existing schemes between inflammatory cells tumor cells etc. based on extensive parameters.

·       5. Word spacing as per template is required at line 175 on page 5.

·       6. The first paragraph on page 7 requires a valid reference.

·       7. The author should refine the paragraph starting from line 76 and discuss it in more detail.

·       8. Why is the paragraph composed of just 2 lines starting from line 74?

·       9. Some of the discussion is without valid references. For example, line 506 on page 13.

·       10. The abstract contains only four sentences but length. It should be revised. 11. The sentence should be a maximum of 2 lines.

·       12. The conclusion should be revised.

·       13. The overall structure of the paper should be revised. Sections 2, 3, 4… should be background, comparison, etc.

Author Response

RE:

Dear reviewer thank for your suggestions.

  1. The referencing protocol has been harmonized.
  2. Sections have been renamed, and the structure of the paper has been partially revised according to the indications for authors given by the journal and referring to the features of the other reviews published in this specific special issue.
  3. The acronym has been corrected.
  4. Dear reviewer thank for your suggestion. Please specify what do you mean for “existing schemes between inflammatory cells tumor cells etc.” and “extensive parameters”. We believe that the different biomarkers provide various information and are detected with very different techniques, so a specific section on their comparison may be confunding.
  5. The entire document has been revised and the correct word spacing has been provided.
  6. The references have been revised in page 7.
  7. The paragraph starting from line 76 has been refined with new added details.
  8. The paragraphs have been revised.
  9. We revised the references. 
  10. The abstract syntax has been corrected
  11. Length of sentences have been revised
  12. Conclusion has been revised
  13. Please, see point 2.

Reviewer 2 Report

In this review, the authors describe studies investigating the possible use of noninvasive tools, including the inflammatory circulating cells, circulating tumor cells (CTCs), cell free DNA (cfDNA), circulating tumor DNA (ctDNA), to provide a more enlightened approach to escalation/deescalation strategies. in early breast cancer. The review is quite informative, but it does not describe how the selection of studies that are included in the review was carried out. You need to add a Methods section. I would also like to see more specific data on the relationship between circulating tumor cells and survival rates, in particular, table 1 should be expanded and the corresponding values should be given. The authors give only the values of p. According to the text of the manuscript, I would also like to see specific odds ratios, if possible.

Author Response

RE: Dear reviewer thank for your suggestions.

We added a Method section and integrated missing data in Table 1 as suggested.

Round 2

Reviewer 2 Report

I have no more comments on the article. I believe that in its present form the manuscript can be recommended for publication.